# Supplementation with Tributyrin for Gestating Sows Reduces Stillborn Rate and Increases Litter Birth Weight

**DOI:** 10.3390/vetsci12030260

**Published:** 2025-03-11

**Authors:** Flávio de Aguiar Coelho, Ana Ligia Braga Mezzina, Ana Clara Rodrigues de Oliveira, Laya Kannan Silva Alves, Nadia de Almeida Ciriaco Gomes, Jorge Yair Perez-Palencia, Marli Arena Dionizio, Eduardo Machado Costa Lima, Soraia Viana Ferreira, Cesar Augusto Pospissil Garbossa

**Affiliations:** 1Department of Nutrition and Animal Production, School of Veterinary Medicine and Animal Science, University of São Paulo, Av. Duque de Caxias Norte, 225, Jardim Elite, Pirassununga 13635-900, São Paulo, Brazil; flavioaguarcoelho@usp.br (F.d.A.C.); anamezzina@usp.br (A.L.B.M.); anaclara0904@usp.br (A.C.R.d.O.); layakannan@usp.br (L.K.S.A.); nadiaciriaco@usp.br (N.d.A.C.G.); 2Department of Animal Science, College of Agriculture, Food and Environmental Sciences, South Dakota State University, Brookings, SD 57007, USA; jorge.perezpalencia@sdstate.edu; 3Perstorp Holding AB, Neptunigatan, 21120 Malmö, Sweden; marli.dionizio@perstorp.com; 4Mcassab—Núcleo de Inovações e Tecnologia, São Paulo 04795-000, São Paulo, Brazil; eduardo.lima@mcassab.com.br; 5DB—Danbred do Brasil, Patos de Minas 38706-000, Minas Gerais, Brazil; soraia@db.agr.br

**Keywords:** butyrate, organic acid, performance, pig nutrition, reproduction, swine production

## Abstract

The search for greater productive efficiency of sows through hyperprolificacy has brought deleterious effects such as long farrowings and less homogeneous litters at birth, as well as a higher incidence of low-weight and low-viability piglets, which can result in fetal lactation and postnatal death. The availability of butyrate, which has important nutritional properties, can benefit the productive performance of pregnant sows by modulating the quality of litters at birth, as well as providing nutritional support for farrowing. This study considered data from one hundred and forty-eight sows supplemented with tributyrin after 35 days of gestation and during lactation. Although the number of piglets born alive and the individual weight of the piglets at birth did not change with the supplementation practiced, the total weight of litters at birth from sows supplemented with tributyrin in the middle and final thirds of the gestational phase was 10.14% higher, and the occurrence of stillborn piglets was reduced by 35.47%. In addition, a trend towards a higher percentage of piglets weighing over 1.4 kg and a lower variation in the weight of the litters from sows supplemented with the organic acid was observed. Therefore, tributyrin supplementation in sows contributes to more homogeneous litters at birth and reduces losses due to stillbirths in prolonged farrowing conditions.

## 1. Introduction

Butyrate glycerides, butyrate salts, and matrix-loaded butyrate have gained significant interest in the swine industry as exogenous sources of butyrate. These compounds possess trophic, anti-inflammatory, antioxidant, and antiapoptotic properties that promote intestinal health and enhance the growth performance of young pigs [1,2,3,4,5,6]. Tributyrin, a triester of butyric acid, falls into this category. Under the action of pancreatic lipase, tributyrin releases a substantial amount of butyrate into the intestinal lumen, supporting cellular energy metabolism, mediating immune response cascades, and preserving transcriptional and gene expression machinery [7].

Advancements in genetic selection to enhance sow prolificacy have resulted in increased variability in individual piglet birth weights, which poses a challenge to piglet survival within the first 72 h of life [8,9]. Primary causes of this variability include genetic, vascular, and autoimmune factors, while secondary causes are often related to inadequate fetal nutrition [10]. In this context, tributyrin supplementation, with its butyrogenic action and role as an energy source, may influence gene expression methylation [11], potentially improving the availability of amino acids and nutrients to foster fetal growth and increase milk production.

Butyrate also plays a critical role in hepatic metabolism. It serves as an important substrate for maintaining both local and peripheral organ health in pigs, not only during fasting but also in the postprandial state [12]. Although the exact amount is yet to be quantified, the fraction of luminal butyrate reaching the liver via the portal vein may contribute to hepatic glycogen storage. This could provide a beneficial glucose substrate, improving sow conditioning for farrowing. Such effects may reduce piglet losses due to perinatal asphyxia [13], result in more vigorous piglets at birth [14], and support faster postpartum recovery in sows [15].

Research on butyrate supplementation in sow nutrition remains limited, but the available literature suggests potential benefits when provided to gestating and lactating sows [15,16,17]. This study hypothesizes that butyrate supplementation may enhance nutritional support to fetuses during gestation, regulate glycemic levels to facilitate farrowing, promote the development of more homogeneous and vigorous litters at birth, and contribute to milk production, leading to heavier and larger litters at weaning.

Thus, the objective of this study was to evaluate the performance of gestating and lactating sows supplemented with tributyrin (1 g/kg of feed) from 35 days of gestation until the end of lactation. Additionally, this study aimed to assess the effects of this supplementation on the performance of piglets during the nursery phase.

## 2. Materials and Methods

The experimental procedures followed the guidelines approved by the Institutional Animal Care and Use Committee (CEUA) of the School of Veterinary Medicine and Animal Science, University of São Paulo, Brazil, under protocol no. 119823112.

### 2.1. Location, Animals, and Housing

The study was conducted from February to September of 2023 on a commercial farrow-to-nursery farm located in the western region of Minas Gerais state, Brazil (18°31′ S; 46°26′ W; 940.3 m altitude), within the Cerrado biome. The farm housed a registered herd of 2800 high-genetic-potential sows.

The experiment included 148 gestating sows of a commercial lineage (DB90, DanBred, Patos de Minas, Brazil), ranging from first to fourth parity, with an average body weight of 201.7 ± 12.2 kg, along with 1640 suckling piglets and 180 weaned piglets. At 35 days of gestation, sows were weighed using a mobile digital scale and housed individually in gestation stalls with solid floors, nipple drinkers, and automated feeders. Sows remained in these stalls until 112 days of gestation.

Sows were then transferred to farrowing rooms, where they were housed in individual farrowing crates with slatted floors, nipple drinkers, and manual feeders, remaining there until weaning at 21 days of lactation. At weaning, piglets were identified, weighed using a digital scale, and transported to the nursery. In the nursery, piglets were grouped into pens of four intact males per pen (average weight: 6.8 ± 0.40 kg). Nursery pens were constructed of masonry with slatted plastic floors, nipple drinkers, and semi-automatic feeders.

During gestation and lactation, sows and piglets were housed in climate-controlled facilities to maintain thermal comfort conditions (18–22 °C for gestating and lactating sows). Auxiliary heated boxes were provided for suckling piglets. Climate control in the nursery was managed with side curtain adjustments and forced ventilation.

On the first day after farrowing, the litter size was standardized separately for each experimental group by adjusting the number of piglets to match the number of viable teats, with one additional piglet per sow above teat capacity. A standardized health protocol was applied to all animals, following the farm’s production unit practices.

### 2.2. Experimental Design

A randomized block design with two treatments was used during gestation and lactation: a control diet without tributyrin supplementation and a diet supplemented with 1 g/kg of tributyrin from a commercial product.

In the nursery phase, a 2 × 2 factorial arrangement was applied, comprising four experimental groups: (i) piglets from tributyrin-supplemented sows supplemented with tributyrin in the nursery phase (tributyrin–tributyrin); (ii) piglets from tributyrin-supplemented sows but not supplemented with tributyrin in the nursery phase (tributyrin–control); (iii) piglets from non-supplemented sows but supplemented with tributyrin in the nursery phase (control–tributyrin); and (iv) piglets from non-supplemented sows with no tributyrin supplementation in the nursery phase (control–control).

Blocking factors for the gestation and lactation stages included sows with 1 to 4 previous parities, the total number of offspring born in the most recent parity, and the sow’s body weight at 35 days of gestation. For the nursery phase, blocking was based on piglet weaning weight and spatial distribution within the nursery facility. This approach aimed to control variability and ensure a balanced allocation of experimental conditions across all groups.

### 2.3. Feeding Protocol

The diet formulation and feeding protocol was designed to meet the nutritional requirements for the experimental phases, following the guidelines outlined by the National Research Council [18] (Table 1 and Table 2). During gestation, feed was provided once daily at 7:00 AM in a restricted manner: 1.8 kg of feed per day from days 35 to 80 of gestation, and 3.0 kg per day from days 81 to 113. Throughout the 21 days of lactation, sows were fed ad libitum, with feed provided five times daily (at 7:00 A.M., 10:30 A.M., 1:00 P.M., 4:30 P.M., and 8:30 P.M.). In the nursery phase, feeding was also ad libitum and divided into three nutritional phases, lasting 15, 14, and 10 days, respectively. Diets in this phase were formulated to include decreasing levels of blood plasma and dairy-derived ingredients.

Tributyrin (ProPhorce™ SR 130, MCassab, São Paulo, Brazil) was incorporated into the gestation and lactation diets at a rate of 0.1%, corresponding to 630 mg of tributyrin (500 mg of butyrate per kg of feed) (Table 3). In the nursery phases, pre-starter, starter 1, and starter 2 diets included tributyrin at inclusion rates of 0.15%, 0.10%, and 0.10%, respectively (Table 4).

During gestation, tributyrin supplementation was provided individually by adding precise amounts directly to each sow’s feeder. The product was weighed using a high-precision scale and stored in 50 mL screw-top plastic containers. For the lactation and nursery phases, tributyrin was mixed into the diets and processed at the feed mill to ensure uniform distribution.

### 2.4. Analyzed Variables

#### 2.4.1. Body Composition and Sow Productive Parameters

To evaluate the mobilization of body reserves in response to the supplementation, sows were weighed at days 35 and 112 of gestation and at weaning. Measurements of backfat thickness and loin depth were taken using an ultrasound device (Model KX5600 VET Chison, Jiangsu, China, equipped with a linear transducer operating at 2.5 MHz) at the P2 position [19] on days 80 and 112 of gestation and at weaning.

Reproductive performance parameters were assessed by recording the start and end times of farrowing, the birth time of each piglet, the number of live-born piglets, stillborns, and mummies. Additionally, individual piglet weights were recorded immediately after birth, during litter equalization, and at weaning. Sow feed intake during lactation was also documented. The start and end times of farrowings were defined as the times of birth for the first and last piglets, respectively.

Blood samples were collected from sows (a drop of blood in the ear vessels) during farrowing at the time of birth of the 1st, 7th, 14th and last piglet, to measure blood glucose concentrations using a portable glucometer (Accu-Chek Guide Meter™, Roche Diabetes Care, Inc., São Paulo, Brazil), following the protocol described by Carnevale et al. [20].

Recorded data were used to calculate farrowing duration, mean birth interval, proportion of farrowing-related losses (stillborn piglets), mummified piglets, and litter daily weight gain (PGDI) during lactation. Sow milk production was estimated using Noblet and Etienne [21] (Equation (1)):Milk production (g/piglet/day) = 2.50 × average piglet gain (g) + 80.2 × piglet body weight at the beginning (kg) + 7(1)

To characterize litter variability, live-born piglets were grouped into the following birth weight categories: (i) less than 0.8 kg, (ii) 0.8–1.0 kg, (iii) 1.0–1.2 kg, (iv) 1.2–1.4 kg, and (v) 1.4 kg or more. At weaning, piglets were classified into the following weight categories: (i) less than 5.5 kg, (ii) 5.5–6.0 kg, (iii) 6.0–6.5 kg, (iv) 6.5–7.0 kg, and (v) 7.0 kg or more.

#### 2.4.2. Performance Parameters of Piglets in the Nursery Phase

During the nursery phase, piglets were individually weighed on days 1, 15, 29, and 39, which corresponded to the nutritional phases of pre-starter (1–15 days in nursery), starter 1 (16–29 days in nursery), and starter 2 (30–39 days in nursery). The average daily weight gain (ADG) was calculated by subtracting the piglet weight from the previous weighing phase and adjusted to the arithmetic mean in each experimental unit. Average daily feed intake (ADI) was determined by subtracting the remaining feed intake from the total amount fed during each phase. The feed conversion rate (FCR) was calculated as the ratio between the average feed intake and weight gain.

### 2.5. Statistical Analysis

All data were tested for normality using the Shapiro–Wilk test. Variables that did not follow a normal distribution were transformed using the RANK procedure of SAS (SAS Institute Inc., Cary, NC, USA) with residual normalization. The effects of treatments and interactions were analyzed using ANOVA with the MIXED procedure of SAS. Models included random effects for parity order, the total number of piglets born in the previous litter, and the sow’s body weight at day 35 of gestation (Equation (2)), as well as the piglet’s weight at weaning and the spatial distribution of piglets in the nursery facility (Equation (3)). All data were described by LSMEANS, and the largest standard error of the mean (SEM) for each variable was presented. Differences between means were considered statistically significant when *p* < 0.05, with trends considered at the 10% level (0.05 < *p* ≤ 0.10). Comparisons were made using the F-test.Y_kij_ = μ + T_k_+ δ_j_ + ε_kj_(2)
where $Y_{ij}$ is the observation in unit j under treatment i; μ is the overall mean; $T_{k}$ is the effect of tributyrin; $\delta_{j}$ is the effect of block j; and $\varepsilon_{kj}$ is the error associated with the observation in block j under treatment k.Y_kij_ = µ + TR_k_ + TN_i_ + (TRTN)_ki_ + δ_j_ + ε_kij_
(3)
where: $Y_{ij}$ is the observation in unit j under treatment i; μ is the overall mean; ${TR}_{k}$ is the effect of tributyrin during reproduction (gestation and lactation); ${TN}_{i}$ is the effect of tributyrin during the nursery phase; ${TRTN}_{ki}$ is the effect of treatments interactions; $\delta_{j}$ is the effect of block j; and $\varepsilon_{kij}$ is the error associated with the observation in block j under treatment ki.

## 3. Results

Of the 148 sows initially included in the study, 47 were excluded due to reasons such as inability to weigh prior to farrowing, lack of adequate infrastructure for data collection in the farrowing room, and nighttime farrows without supervision. Consequently, the final dataset included 101 sows with complete data collected up to the end of the lactation.

Tributyrin supplementation during gestation and lactation did not influence the body weight parameters or the body composition of the sows (Table 5). However, a trend (*p* = 0.0532) was observed for increased body weight accumulation, with an average increase of 3.05 kg, and a 6.51% reduction (*p* = 0.099) in dorsal fat thickness in the sows supplemented with tributyrin between days 35 and 112 of gestation compared to those that were not supplemented. Blood glucose levels measured throughout parturition were not influenced by the diets provided during the gestational period (Figure 1).

Supplementation with tributyrin during the middle and late stages of gestation reduced stillborn losses by 35.47% (*p* = 0.032), without affecting farrowing duration or the occurrence of mummified piglets (Table 6). The litter weight at birth was 10.14% higher (*p* = 0.018) in sows consuming tributyrin; however, no effects were observed on individual piglet birth weight. The proportion of piglets with birth weights between 1.0 and 1.2 kg was 33.17% higher in sows that were not supplemented with tributyrin (*p* = 0.023). Nonetheless, supplementation with tributyrin tended to reduce (*p* = 0.053) the coefficient of variation in litter weight by 18.73% and increase (*p* = 0.057) the proportion of piglets weighing 1.4 kg or more by 27.85%.

The performance of sows and their litters during lactation was not affected by tributyrin supplementation during the gestation and lactation phases (Table 7). Sows’ feed intake during lactation and their estimated milk production were similar for both treatments. Likewise, the productive development of the litters during lactation and the coefficient of variation at weaning were not influenced by tributyrin supplementation.

No interaction effect was observed between supplementation with tributyrin during the gestation and lactation phases and the nursery phase for any of the analyzed variables (Table 8). Piglets from sows supplemented with tributyrin during gestation and lactation tended (*p* = 0.056) to have a higher ADFI by 28 g during the first 15 days of the nursery phase. However, supplementation with tributyrin during the nursery phase negatively affected the FCR of piglets between 15 and 29 days of the nursery phase (*p* = 0.035), worsening it by 3.16%. This effect tended (*p* = 0.099) to persist, with a 3.55% worse feed conversion by the end of the nursery phase.

## 4. Discussion

The accumulation of body reserves during gestation is essential for supporting the nutritional needs of sows through to the end of lactation [22]. Triglyceride hydrolysis may serve as a glycogenic substrate, especially during the intensification of myometrial contractions during farrowing [14] and throughout lactation [23]. Butyrate plays a significant role in promoting adipogenesis, lipogenesis, and the expression of adipokines in preadipocyte cells from pigs treated in vitro [24], potentially enhancing energy reserves in pregnant sows, making them available during periods of greater demand. Supplementation with tributyrin (250 mg/kg of butyrate in the diet) during the final 35 days of gestation has been shown to reduce the duration of parturition, suggesting that tributyrin can provide sufficient energy to meet nutritional demands during this critical period [15].

However, our results indicated a reduction in backfat accumulation in sows prepartum when supplemented with tributyrin, which is consistent with findings from Cooper [25], who reported that prenatal exposure to tributyrin at a 2% dietary level reduced backfat accumulation in pregnant sows. Butyrate, an agent influencing nutrient partitioning in adipose tissue, may regulate genes involved in fatty acid oxidation, reduce lipogenesis, and enhance lipolysis [26]. Additionally, butyrate modulates the hepatic metabolic profile in pigs fed this fatty acid [27,28], with the potential to reduce losses associated with the gestational process.

Prepartum weight gain in sows was associated with increased litter weights, which showed a 2 kg increase at birth. This is consistent with the findings of [17], who reported that sows supplemented with tributyrin during gestation tended to have heavier litters at birth, although individual piglet birth weights were unaffected. This effect is likely due to butyrate’s action on the lipolytic pathway, which directs more nutrients to the placenta, improving fetal nutrition and contributing to a reduction in stillborns—a common occurrence in larger litters [29,30]—and low birth weight [31]. These responses are characteristic of the hyperprolific genetics used in the study, with litters exceeding 17 piglets [32], where butyrate supplementation played a key role in reducing reproductive losses in sows.

Another important factor in hyperprolificity is the heterogeneity of birth weights, which is closely related to larger litters. In addition to the impact of nutrient distribution among a greater number of fetuses [9], the proportion of fetuses located farther from the ovarian end also increases [33]. These regions, which receive lower blood flow, result in lighter piglets at birth.

The trend of reduced variability in piglet birth weight observed in our study can be attributed to both the greater nutrient availability during the middle and late stage of gestation and the activation of histone acetylation. This process intensifies gene transcription regulation involved in the early development of offspring tissues, as supplementation with tributyrin began at 35 days of gestation. Supplementation with butyrate during pregnancy may enhance adipogenesis and increase fat accumulation in muscle tissue, likely mediated by hyperacetylation in the promoter regions of lipogenic genes, as observed in studies with rats and neonatal piglets [34,35]. However, further research is necessary to fully elucidate the mechanisms behind this effect. A possible hypothesis is that maternal supplementation with butyrate during the middle and late stages of gestation may regulate gene transcription machinery, modulating the development of primary tissues, influencing fetal uterine growth, and ultimately resulting in piglets with greater vitality at birth.

Maternal supplementation can influence offspring growth performance throughout their lives. However, by the end of lactation, weaning weight and the weight variation among piglets within the litter were not influenced by maternal supplementation during gestation and lactation. This finding contrasts with previous studies that reported improved weaning performance in piglets [15] and finisher pigs [26] born to sows supplemented with tributyrin during gestation and lactation. Despite this, our study did show a trend toward increased feed intake by piglets during the first 15 days post-weaning. This supports the hypothesis that offspring may develop a greater preference for ingredients provided to their mothers during gestation [36,37] due to stimuli that enhance the acceptability of flavors in the progeny’s later life. Piglets exposed to certain flavors during prenatal life tend to prefer those same flavors during lactation and show increased feed intake when those flavors are incorporated into their post-weaning diets [38,39]. Therefore, although maternal supplementation did not demonstrate significant benefits on piglet performance during the nursery phase, the increased feed intake observed during the first 15 days post-weaning suggests that tributyrin may help mitigate post-weaning food neophobia in piglets.

Butyrate-loaded molecules adhered to glycerol have the potential to release free butyrate in post-gastric portions when acted upon by lipase enzymes [40]. This process can favor pH modulation and the nutrition of colonic cells, promoting health and reducing stress associated with early weaning, which can impact the performance and welfare of young piglets [41,42]. In this study, supplementation with tributyrin during the nursery phase did not improve performance, which aligns with findings from Barbosa et al. [43], who reported no performance improvements despite better intestinal function in weaned piglets. However, the inclusion of 0.2% tributyrin in nursery diets resulted in improved body weight, ADG, and feed conversion, which contradicts the poorer feed conversion observed in our study. The divergence in the results may be attributed to the higher dosage (0.2%) used in that study, which was double the amount utilized in the present study.

## 5. Conclusions

The supplementation of tributyrin during gestation reduced the occurrence of stillborn piglets and increased litter weight at birth. However, performance during lactation and the nursery phase was not influenced by tributyrin supplementation during the gestation, lactation, and nursery phases. Therefore, the results of this study suggest that including tributyrin in the diet of gestating sows is a strategy to improve performance and reduce losses associated with modern highly prolific lines.

## Figures and Tables

**Figure 1 vetsci-12-00260-f001:**
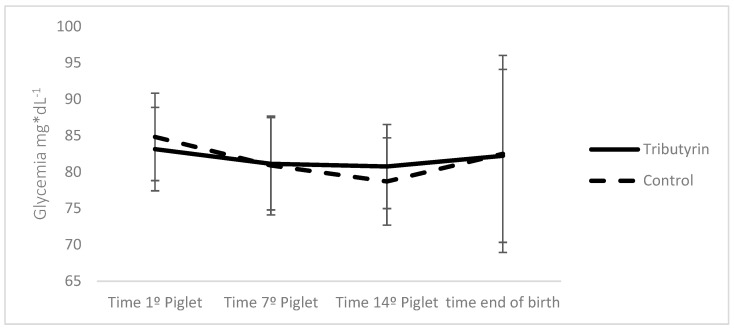
Average glycemic values during farrowing of sows supplemented or not supplemented with tributyrin during the gestational phase.

**Table 1 vetsci-12-00260-t001:** Calculated nutritional composition of diets fed to sows during gestation and lactation.

Diets	Gestation		Lactation	
Treatments	Control	Tributyrin	Control	Tributyrin
Metabolizable energy, kcal/kg	3210	3210	3500	3500
Crude protein, %	14.58	14.58	20.92	20.92
Fat, %	2.52	2.52	6.88	6.88
Crude fiber, %	3.04	3.04	2.97	2.97
Ash, %	4.82	4.82	10.15	10.15
Total calcium, %	0.75	0.75	0.85	0.85
Total phosphorus, %	0.55	0.55	0.75	0.75
Available phosphorus, %	0.45	0.45	0.56	0.56
Digestible lysine, %	0.60	0.60	1.17	1.17
Digestible methionine + cystine, %	0.43	0.43	0.70	0.70
Digestible threonine, %	0.47	0.47	0.82	0.82
Digestible tryptophan, %	0.16	0.16	0.28	0.28
Sodium, %	0.22	0.22	0.28	0.28
Chlorine, %	0.37	0.37	0.40	0.40
Iron, mg/kg	80.00	80.00	67.20	67.20
Copper, mg/kg	37.50	37.50	87.50	87.50
Manganese, mg/kg	40.00	40.00	33.60	33.60
Zinc, mg/kg	168.40	168.40	92.40	92.40
Cobalt, mg/kg	0.20	0.20	0.17	0.17
Iodine, mg/kg	1.40	1.40	1.21	1.21
Chrome, mg/kg	-	-	0.40	0.40
Selenium, mg/kg	0.45	0.45	0.35	0.35
Vitamin A, UI/g	13.50	13.50	12.00	12.00
Vitamin D3 (Cholecalciferol), UI/g	2.81	2.81	2.50	2.50
Vitamin E, UI/kg	78.75	78.75	70.00	70.00
Vitamin K3, mg/kg	2.81	2.81	2.50	2.50
Vitamin B1 (Thiamine), mg/kg	2.47	2.47	2.20	2.20
Vitamin B2 (Riboflavin), mg/kg	9.00	9.00	8.00	8.00
Vitamin B6 (Pyridoxine), mg/kg	3.37	3.37	3.00	3.00
Vitamin B12, Mcg/kg	33.75	33.75	30.00	30.00
Niacin, mg/kg	33.75	33.75	30.00	30.00
Pantothenic acid, mg/kg	22.50	22.50	20.00	20.00
Folic acid, mg/kg	3.37	3.37	3.00	3.00
Biotin, mg/kg	0.85	0.85	0.82	0.82
Choline, mg/kg	1.93	1.93	1.89	1.89
Butyrate, mg/kg	-	0.50	-	0.50

**Table 2 vetsci-12-00260-t002:** Calculated nutritional composition of diets fed to piglets in the nursery phase, from sows supplemented or not supplemented with tributyrin during gestation and lactation.

Calculated Values	Control	Tributyrin	Control	Tributyrin	Control	Tributyrin
Diets	Pre-Starter		Starter 1		Starter 2	
Metabolizable energy, kcal/kg	3.54	3.54	3.55	3.55	3.43	3.43
Crude protein, %	21.13	21.13	20.81	20.81	19.29	19.29
Fat, %	5.08	5.08	4.92	4.92	5.22	5.22
Ash, %	7.65	7.65	7.92	7.92	7.65	7.65
Total phosphorus, %	0.69	0.69	0.68	0.68	0.49	0.49
Available phosphorus, %	0.55	0.55	0.50	0.50	0.40	0.40
Total calcium, %	0.79	0.79	0.90	0.90	0.70	0.70
Digestible valine, %	1.01	1.01	0.95	0.95	0.80	0.80
Digestible lysine, %	1.46	1.46	1.33	1.33	1.22	1.22
Digestible methionine + cystine, %	0.84	0.84	0.74	0.74	0.70	0.70
Digestible threonine, %	1.01	1.01	0.88	0.88	0.80	0.80
Digestible tryptophan, %	0.33	0.33	0.30	0.30	0.25	0.25
Sodium, %	0.32	0.32	0.25	0.25	0.24	0.24
Iron, mg/kg	82.99	82.99	82.99	82.99	80.04	80.04
Copper, mg/kg	150.00	150.00	125.25	125.25	124.99	124.99
Manganese, mg/kg	40.00	40.00	40.00	40.00	40.02	40.02
Zinc, mg/kg	2400.00	2400.00	1934.92	1934.92	1205.05	1205.05
Cobalt, mg/kg	0.20	0.20	0.20	0.20	0.20	0.20
Iodine, mg/kg	1.33	1.33	1.33	1.33	1.40	1.40
Selenium, mg/kg	0.35	0.35	0.35	0.35	0.35	0.35
Vitamin A, UI/g	15.03	15.03	15.03	15.03	9.00	9.00
Vitamin D3 (Cholecalciferol), UI/g	3.13	3.13	3.13	3.13	1.88	1.88
Vitamin E, UI/kg	87.68	87.68	87.68	87.68	52.50	52.50
Vitamin K3, mg/kg	3.13	3.13	3.13	3.13	1.88	1.88
Vitamin B1 (Thiamine), mg/kg	2.76	2.76	2.76	2.76	1.65	1.65
Vitamin B2 (Riboflavin), mg/kg	10.02	10.02	10.02	10.02	6.00	6.00
Vitamin B6 (Pyridoxine), mg/kg	3.76	3.76	3.76	3.76	2.25	2.25
Vitamin B12, Mcg/kg	37.58	37.58	37.58	37.58	22.50	22.50
Niacin, mg/kg	37.58	37.58	37.58	37.58	22.50	22.50
Pantothenic acid, mg/kg	25.05	25.05	25.05	25.05	15.00	15.00
Folic acid, mg/kg	3.76	3.76	3.76	3.76	2.25	2.25
Biotin, mg/kg	0.50	0.50	0.50	0.50	0.34	0.34
Choline, mg/kg	1.63	1.63	1.63	1.63	1.82	1.82
Milk Protein, %	3.98	3.98	1.85	1.85	-	-
Lactose, %	12.97	12.97	5.58	5.58	-	-
Butyrate, mg/kg	-	750.00	-	500.00	-	500.00

**Table 3 vetsci-12-00260-t003:** Centesimal composition of diets used for sows during gestation and lactation.

Ingredients, kg	Gestation		Lactation	
	Control	Tributyrin	Control	Tributyrin
Corn	73.98	73.98	50.83	50.83
Corn Gluten	-	-	2.53	2.53
Soybean Meal	18.93	18.93	31.73	31.73
Meat and Bone Meal	-	-	3.00	3.00
Degummed Soybean Oil	-	-	4.53	4.53
Sugar	-	-	4.00	4.00
FIBERMILL ^1^	4.00	4.00	-	-
Enzyme ^5^	-	-	0.005	0.005
Yeast Wall ^2^	0.08	0.08	-	-
Prebiotic + Probiotic ^2^	0.01	0.01	0.02	0.02
Limestone	0.63	0.63	-	-
Dicalcium Phosphate	1.23	1.23	1.25	1.25
Adsorbent	0.05	0.05	0.05	0.05
Iodized Salt	0.50	0.50	0.50	0.50
Sodium Bicarbonate	-	-	0.13	0.13
Chromium	-	-	0.01	0.01
Organic Selenium	-	-	0.01	0.01
Vitamin C	0.03	0.03	0.15	0.15
Reproduction Pig Nucleus ^3^	0.50	0.50	-	-
Lactation Pig Nucleus ^4^	-	-	0.50	0.50
Flavoring	-	-	0.04	0.04
L-Lysine	-	-	0.25	0.25
DL-Methionine	0.03	0.03	0.16	0.16
L-Threonine	0.03	0.03	0.17	0.17
L-Tryptophan	-	-	0.05	0.05
L-Valine	-	-	0.08	0.08
L-Carnitine	-	-	0.002	0.002
Tributyrin ^6^	-	0.10	-	0.10

^1^ Commercial product composed of soluble and insoluble fibers, 12% crude fiber; ^2^ product composed of yeast wall; ^3^ vitamin and mineral premix composition, supplied per kg of gestation diet, SUI Rep Feed, SUINCO, Brazil: iodine 0.2 mg/kg; copper 17 mg/kg; iron 13 mg/kg; manganese 16.7 mg/kg; selenium 0.055 mg/kg; zinc 18 mg/kg; cobalt 0.03 mg/kg; pantothenic acid 3.99 mg/kg; folic acid 0.59 mg/kg; biotin 0.16 mg/kg; niacin 5.9 mg/kg; vitamin A 2 390 IU/g; vitamin B1 0.43 mg/kg; vitamin B2 1.59 mg/kg; vitamin B6 0.59 mg/kg; vitamin B12 5.9 mcg/kg; vitamin D3 490 IU/g; vitamin E 13 IU/kg, vitamin K3 0.49 mg/kg. ^4^ Vitamin and mineral premix composition, supplied per kg of lactation diet, SUI Lac Feed, SUINCO, Brazil: iodine 0.23 mg/kg; copper 75 mg/kg; iron 15 mg/kg; manganese 7.9 mg/kg; selenium 0.065 mg/kg; zinc 33 mg/kg; cobalt 0.038 mg/kg; pantothenic acid 4.99 mg/kg; folic acid 0.67 mg/kg; biotin 0.165 mg/kg; niacin 6.74 mg/kg; vitamin A 2 690 IU/g; vitamin B1 0.49 mg/kg; vitamin B2 1.79 mg/kg; vitamin B6 0.67 mg/kg; vitamin B12 6.74 mcg/kg; vitamin D3 560 IU/g; vitamin E 15.74 IU/kg, vitamin K3 0.56 mg/kg. ^5^ Carbohydrase, 500 mg/kg; ^6^ Tributyrin, ProPhorce™ SR 130, MCassab, Brazil: 1 g/g (500 mg of butyrate).

**Table 4 vetsci-12-00260-t004:** Centesimal composition of diets used in the nursery phase for piglets from sows supplemented or not supplemented with tributyrin during gestation and lactation.

Component, kg	Pre-Starter		Starter 1		Starter 2	
	Control	Tributyrin	Control	Tributyrin	Control	Tributyrin
Corn	31.21	31.21	41.51	41.51	59.46	59.46
Soybean Meal	10.07	10.07	21.07	21.07	30.00	30.00
CP24 ^1^	20.00	20.00	13.33	13.33	-	-
Spray-dried animal plasma	4.00	4.00	1.50	1.50	-	-
Sugar	5.00	5.00	5.00	5.00	4.00	4.00
Dried whey	12.50	12.50	5.00	5.00	-	-
Lactose ^2^	10.00	10.00	5.00	5.00	-	-
Degummed Soybean Oil	1.50	1.50	2.00	2.00	2.67	2.67
Dicalcium Phosphate	1.62	1.62	1.58	1.58	0.83	0.83
Limestone	-	-	-	-	0.65	0.65
Inert	0.20	0.05	0.20	0.10	0.10	-
Salt	0.10	0.10	0.30	0.30	0.50	0.50
Fumaric Acid	0.50	0.50	-	-	-	-
Enzyme Carbohydrase	-	-	0.005	0.005	0.005	0.005
Ultracid ^4^	0.30	0.30	0.50	0.50	0.30	0.30
Mineral + vitamin premix ^3^	3.00	3.00	3.00	3.00	1.50	1.50
Tributyrin ^5^	-	0.15	-	0.10	-	0.10

^1^ CP24: dry matter, 88.7%; crude protein, 27.12%; fat, 0.95%; crude fiber, 1.42%; ash, 2.60%; digestible calcium., 0.134%; available phosphorus., 0.134%; total digestible nutrients, 52.65%; amido, 48.1%; metabolizable energy, 3.541 Mcal/kg; digestible valine, 1.37%; total lysine, 1.436%; methionine + cystine, 0.789%; threonine, 0.946%; tryptophan, 0.321%; total leucine, 2.154%; total isoleucine, 1.243%; sodium, 0.031%; chlorine, 0.026%; potassium, 0.763%; magnesium, 0.072%; sulfur, 0.065%; choline, 0.065 g/kg. ^2^ Lactose-based product; ^3^ supplied per kilogram of diet: vitamin A, 2,200 IU; vitamin D3, 220 IU; vitamin E, 16 IU; vitamin K, 0.5 mg; thiamine, 1.5 mg; riboflavin, 4 mg; niacin, 30 mg; pantothenic acid, 12 mg; vitamin B12, 0.02 mg; folic acid, 0.3 mg; Cu, 6 mg as copper sulfate; I, 0.14 mg as calcium iodate; Fe, 100 mg as ferrous sulfate; Mn, 4 mg as manganese oxide; Se, 0.3 mg as sodium selenite; Zn, 100 mg as zinc oxide; biotin 0.2 mg; ^4^ organic acid; ^5^ tributyrin, ProPhorce™ SR 130, MCassab, Brazil: 1 g/g (500 mg of butyrate).

**Table 5 vetsci-12-00260-t005:** Average values of body weight and body composition of sows supplemented or not supplemented with tributyrin during gestation and lactation.

Item	Treatment		SEM ^1^	*p*-Value ^2^
	Control	Tributyrin		
Number of Sows	48	53	-	-
Gestation				
Initial body weight day 35, kg	202.75	199.97	12.207	0.836
Body weight day 112, kg	262.75	259.32	12.730	0.474
Body weight change, kg	49.82	52.87	7.020	0.053
Backfat thickness, mm	16.60	15.52	1.140	0.099
Loin depth, mm	50.13	49.61	1.730	0.710
Lactation				
Body weight at weaning, kg	242.96	238.14	10.480	0.277
Body weight change, kg	−21.74	−22.62	2.910	0.288
Backfat thickness, mm	14.79	13.65	0.820	0.107
Backfat thickness change, %	−10.64	−11.78	3.090	0.515
Loin depth, mm	50.86	49.13	1.820	0.130
Loin depth change, %	−3.59	−1.93	2.170	0.487

^1^ Maximum value of standard error of the means; ^2^ means differ statistically using the F-test at a 5% level of significance.

**Table 6 vetsci-12-00260-t006:** Average values of farrowing parameters and litter weight distribution of sows supplemented or not supplemented with tributyrin during the middle and final thirds of gestation.

Item	Treatment		SEM ^1^	*p*-Value ^2^
	Control	Tributyrin		
Number of sows	48	53	-	-
Farrowing duration, min	298.86	286.13	22.668	0.737
Average birth interval, min	17.17	16.46	1.387	0.679
Total born, piglets per litter	17.48	17.69	0.442	0.763
Liveborn, piglets per litter	15.49	16.27	0.585	0.199
Mummies, %	4.52	4.03	1.382	0.661
Stillborn, %	6.54	4.22	1.133	0.032
Litter weight distribution at birth				
Piglet weight, kg	1.31	1.37	0.111	0.109
Litter weight, kg	20.01	22.04	1.220	0.018
Piglet < 0.8 kg, %	6.51	4.92	2.227	0.260
Piglet ≥ 0.8–<1.0 kg, %	12.55	8.71	3.350	0.103
Piglet ≥ 1.0–<1.2 kg, %	19.43	14.59	3.432	0.023
Piglet ≥ 1.2–<1.4 kg, %	24.35	23.85	2.630	0.864
Piglet ≥ 1.4 kg, %	37.59	48.06	11.142	0.057
Coefficient of variation of litter weight, %	23.76	19.31	2.367	0.053

^1^ Maximum value of standard error of the means; ^2^ means differ statistically using the F-test at a 5% level of significance.

**Table 7 vetsci-12-00260-t007:** Average values of performance parameters of lactating sows supplemented or not supplemented with tributyrin during gestation and lactation phases.

Item	Treatment		SEM ^1^	*p*-Value ^2^
	Control	Tributyrin		
Number of sows	53	48	-	-
Duration of lactation, days	21	21	-	-
Total feed consumption, kg	140.31	143.21	11.893	0.440
Daily feed consumption, kg	6.68	6.82	0.566	0.439
Milk production, kg × Days^−1^	9.83	10.14	11.895	0.537
Litter profile				
Piglet/sow at equalization, N	14.28	14.87	0.265	0.222
Piglet weight at equalization, kg	1.36	1.40	0.073	0.248
Coefficient of variation of litter weight at equalization, %	14.68	14.96	0.701	0.674
Total weaned, piglets per litter	11.87	12.26	0.540	0.262
Piglet weight at weaning, kg	6.12	6.10	0.424	0.874
Litter weight at weaning, kg	73.31	74.93	5.127	0.502
Daily weight gain of the litter, kg	2.58	2.57	0.214	0.871
Litter weight distribution at weaning				
Piglet < 5.5 kg, %	29.32	28.95	6.747	0.890
Piglet ≥ 5.5 < 6.0 kg, %	12.39	14.85	2.927	0.511
Piglet ≥ 6.0 < 6.5 kg, %	17.80	14.87	2.549	0.370
Piglet ≥ 6.5 < 7.0 kg, %	17.82	14.78	3.010	0.221
Piglet ≥ 7.0 kg, %	22.02	25.63	12.455	0.304
Coefficient of variation of litter weight, %	15.21	15.87	0.921	0.851

^1^ Maximum value of standard error of the means; ^2^ means differ statistically using the F-test at a 5% level of significance.

**Table 8 vetsci-12-00260-t008:** Average values of the productive performance of intact male piglets born from sows supplemented or not supplemented with tributyrin during gestation and lactation phases and supplemented or not with tributyrin during the nursery phase.

Item		Nursery ^1^		Average ^2^	SEM ^3^	*p*-Value ^4^		
		Control	Tributyrin			Reproduction ^2^	Nursery ^1^	R × N ^5^
Phase 1: 15 days								
Body weight—initial, kg	Control	6.79	6.85	6.82	0.409	0.900	0.928	0.970
	Tributyrin	6.76	6.78	6.77				
	Average ^1^	6.77	6.81					
Body weight, kg	Control	11.66	11.84	11.75	0.552	0.620	0.710	0.950
	Tributyrin	11.91	12.15	12.03				
	Average	11.79	11.99					
Average daily feed intake, kg	Control	0.40	0.41	0.41	0.015	0.056	0.232	0.549
	Tributyrin	0.42	0.45	0.43				
	Average	0.41	0.43					
Average daily weight gain, kg	Control	0.32	0.33	0.33	0.016	0.190	0.486	0.842
	Tributyrin	0.34	0.36	0.35				
	Average	0.33	0.35					
Feed-to-gain ratio	Control	1.24	1.24	1.24	0.026	0.755	0.720	0.665
	Tributyrin	1.24	1.26	1.25				
	Average	1.24	1.25					
Phase 2: 14 days								
Body weight, kg	Control	19.73	19.76	19.75	0.752	0.698	0.898	0.934
	Tributyrin	19.96	20.12	20.04				
	Average	19.84	19.94					
Average daily feed intake, kg	Control	0.89	0.91	0.90	0.029	0.967	0.599	0.988
	Tributyrin	0.89	0.91	0.90				
	Average	0.89	0.91					
Average daily weight gain, kg	Control	0.58	0.57	0.57	0.017	0.938	0.648	0.897
	Tributyrin	0.58	0.57	0.57				
	Average	0.58	0.57					
Feed-to-gain ratio	Control	1.55	1.60	1.57	0.022	0.951	0.035	0.928
	Tributyrin	1.55	1.59	1.57				
	Average	1.55	1.60					
Phase 3: 10 days								
Body weight, kg	Control	26.11	25.83	25.97	0.887	0.672	0.966	0.792
	Tributyrin	26.25	26.45	26.35				
	Average	26.18	26.14					
Average daily feed intake, kg	Control	1.13	1.13	1.13	0.036	0.999	0.731	0.674
	Tributyrin	1.12	1.14	1.13				
	Average	1.12	1.14					
Average daily weight gain, kg	Control	0.64	0.61	0.62	0.018	0.648	0.464	0.351
	Tributyrin	0.63	0.63	0.63				
	Average	0.63	0.62					
Feed-to-gain ratio	Control	1.77	1.86	1.82	0.034	0.475	0.099	0.552
	Tributyrin	1.77	1.81	1.79				
	Average	1.77	1.84					

^1^ Average factor: supplementation in the nursery; ^2^ average factor: supplementation in reproduction; ^3^ maximum value of standard error of the means; ^4^ means differ statistically using the F-test at a 5% level of significance; ^5^ interaction between reproduction × nursery factors.

## Data Availability

The raw data supporting the conclusions of this article will be made available by the authors on request.

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
