# Peer review of "Supplementation with Tributyrin for Gestating Sows Reduces Stillborn Rate and Increases Litter Birth Weight"

_vetsci, 2025, doi:10.3390/vetsci12030260_

Round 1
Reviewer 1 Report
Comments and Suggestions for Authors
This study supplemented tributyrin in the diet of gestating and lactating sows and analyzed its effects on the performances. The findings indicate that supplementation with tributyrin can reduce stillborn rate and increase litter birth weight. This indicated the potential of tributyrin supplementation in the diet of gestating sows to improve the performances, which provides a novel strategy to enhance the benefits of pig farming. Several issues are needed to be further addressed.
- Figure 1, Error bars of tributyrin and control should be separately displayed.
- This study revealed that tributyrin supplementation can improve performances of gestating and lactating sows. The potentially specific genes, signaling pathways, metabolic process, or physiological status involved in the benefic effects of tributyrin on the traits studied in this study can be further discussed.
- In recent years, many additives have been found to improve pig performance. I suggest the authors provide potential advantages of butyrate than the known similar additives in the discussion section.
Author Response
This study supplemented tributyrin in the diet of gestating and lactating sows and analyzed its effects on the performances. The findings indicate that supplementation with tributyrin can reduce stillborn rate and increase litter birth weight. This indicated the potential of tributyrin supplementation in the diet of gestating sows to improve the performances, which provides a novel strategy to enhance the benefits of pig farming. Several issues are needed to be further addressed.
- Figure 1, Error bars of tributyrin and control should be separately displayed.
Comment: Thank you for your comment. The adjustments have been made and the new version of the figure has been inserted.
- This study revealed that tributyrin supplementation can improve performances of gestating and lactating sows. The potentially specific genes, signaling pathways, metabolic process, or physiological status involved in the benefic effects of tributyrin on the traits studied in this study can be further discussed.
Comment: Thank you for your comment. The discussion has been improved and is highlighted in the revised version. We would like to state that the discussion is limited to the variables adopted in the study and the possible hypotheses involving the expression of potential genes and the modulation of metabolic pathways that through the action of butyrate are cited in specific studies that we used to collaborate with our findings.
- In recent years, many additives have been found to improve pig performance. I suggest the authors provide potential advantages of butyrate than the known similar additives in the discussion section.
Comment: Thank you for your comment. The particularities of tributyrin compared to other similar molecules are well documented in the introduction. Its potential advantage is due to the fact that it is more loaded with butyrate with greater potential to reach the distal third of the intestine, where it is believed to have greater benefits (citation 7).

Reviewer 2 Report
Comments and Suggestions for Authors
There are some spots in the abstract where there should be spaces around punctuation (i.e. 2 x 2 instead of 2x2, p = 0.053 instead of p=0.053. I assume this was done to get the word count under 200.
In Materials and Methods, is the parity of the sow how many parities she has completed or the parity that she is pregnant with for the study? This needs to be made clear because you say you account for litter size of the previous litter. If it is the parity of the current litter that the sow is pregnant with, how do you account for previous litters on a parity 1 sow which would not have had a previous litter?
You have the calculated nutritional content of the diets but was this what was actually fed? Did you analyze the diets to ensure that they were equal? Since you use gestation and lactation to refer to the diets that the sows received, I would consider using Nursery 1, Nursery 2, and Nursery 3 for the nursery diets instead of pre-starter, starter 1, and starter 2.
In Table 3, double-check the values for L-Threonine in the Lactation diet. Everything else is the same between the two diets but these differ slightly. Also, double-check the values for corn under Starter 2 in Table 4.
Section 2.4.1.:
Line 189: Why were sows weighed on d 112 if they were moved into farrowing on d 110? So you moved the sows into farrowing on d 110 (lines 103-104) and then moved the sows back out to weigh them on d 112?
Lines 199-200: It is not clear if these blood samples are being taken on the sow or the piglet. Please reword to make it clear.
Line 204: Mummies are NOT from parturition-related losses. These are piglets that were lost earlier in the gestation period but after calcification of bone that were being reabsorbed by the sow.
Line 205: Suckling piglet daily weight gain: Is this the average daily weight gain of the piglets in the litter or the litter daily weight gain? I would refrain from using piglet if this is the litter daily weight gain.
Eq. 1: Line 207: What is "(piglet)" after weight gain? Is this the number of piglets? If you weighed all the individual piglets, why not just use the sum of their birth weights rather than an average birth weight ("initial body weight (kg piglet -1)") x the number of piglets?
Lines 209-212: Is there a reason you chose the weight categories you did for birth and weaning?
Lines 216-242: FCR and ADFI had to be calculated on a per pen basis (4 boars/pen according to lines 108-109). Weights were recorded on individual pigs but the treatment is applied to the entire pen. Was ADG during the nursery analyzed as a repeated measures with pen as the subject or was it analyzed on the group of 4 pigs? This is unclear and would be the only way to properly analyze ADG.
Line 231: missing a space - p < 0.05 instead of p <0.05
Tables 5-8: There are several spots where a "," is used instead of a "." to separate the whole number from the tenths/hundredths places.
Table 5: Consider including d35 with initial body weight as a reminder that the initial body weight of the sow was recorded on d35 of gestation.
Also, regarding sow weights: did you consider using the equations by Noblet et al., 1985 (Br. J. Nutr. 53:251-265) to adjust the sow weight for fetal mass and milk production?
Table 6: Total born and live born under Control have only 1 number behind the decimal, while the others have 2. Please be consistent in the usage of decimal places. Same for p-value for Average birth interval. All your other p-values are to 3 decimal places.
Table 6, 7: Did you consider analyzing the number of piglets in each weight category using litter size as a covariate instead of analyzing the percentage of piglets in each weight category?
Table 8: Body weight - initial = have a superscript on Average of 4 which has a footnote that states that the Means differ significantly. However, if I read your table correctly, the difference between these two values has a p-value of 0.9. I assume this should be a superscript of 6 with a footnote that states "Average factor: supplementation in nursery" similar to superscript 2. Also, in the footnotes, superscript 5 is not in superscript format. I would also consider using "sow" instead of "reproduction" for this table.
Lines 328-329: Wording could be better. Also, need citation on blood flow to the uterine horns. You state "typically result in lighter piglets" but reference 33 says it does and you don't have any other citations (i.e. the manuscripts 33 references as having shown differing effects of uterine horn position on birth weight).
Abbreviations: You have abbreviations listed here that I didn't see used in the manuscript and abbreviations that were used in the manuscript that are missing from this list.
Author Response
- There are some spots in the abstract where there should be spaces around punctuation (i.e. 2 x 2 instead of 2x2, p = 0.053 instead of p=0.053. I assume this was done to get the word count under 200.
Comment: Thank you for your comment. The abstract has been adjusted and is highlighted in the revised version.
- In Materials and Methods, is the parity of the sow how many parities she has completed or the parity that she is pregnant with for the study? This needs to be made clear because you say you account for litter size of the previous litter. If it is the parity of the current litter that the sow is pregnant with, how do you account for previous litters on a parity 1 sow which would not have had a previous litter?
Comment: Thank you for your comment. The parity of a sow refers to the number of complete farrowing cycles. Therefore, a sow with parity 1 has already farrowed once.
- You have the calculated nutritional content of the diets but was this what was actually fed? Did you analyze the diets to ensure that they were equal? Since you use gestation and lactation to refer to the diets that the sows received, I would consider using Nursery 1, Nursery 2, and Nursery 3 for the nursery diets instead of pre-starter, starter 1, and starter 2.
Comment: Thank you for your comment. Both treatments received the same CONTROL diet corresponding to each phase of the study, with the inclusion of tributyrin included in the diet above for the set of subjects in the TRIBUTYRIN treatment.
- In Table 3, double-check the values for L-Threonine in the Lactation diet. Everything else is the same between the two diets but these differ slightly. Also, double-check the values for corn under Starter 2 in Table 4.
Comment: Thank you for your comment. The adjustments were made in the new version. The L-Threonine values in the lactation diet were formatted to two decimal places. The values ​​for corn in Starter 2 were a typing error.
- Section 2.4.1.:
Line 189: Why were sows weighed on d 112 if they were moved into farrowing on d 110? So you moved the sows into farrowing on d 110 (lines 103-104) and then moved the sows back out to weigh them on d 112?
Comment: Thank you for your comment. The adjustments were made in the new version. The sows were taken to the maternity ward at 112 days of pregnancy, the date on which they were weighed.
Lines 199-200: It is not clear if these blood samples are being taken on the sow or the piglet. Please reword to make it clear.
Comment: Thank you for your comment. The adjustments were made in the new version.
Line 204: Mummies are NOT from parturition-related losses. These are piglets that were lost earlier in the gestation period but after calcification of bone that were being reabsorbed by the sow.
Comment: Thank you for your comment. The adjustments were made in the new version.
Line 205: Suckling piglet daily weight gain: Is this the average daily weight gain of the piglets in the litter or the litter daily weight gain? I would refrain from using piglet if this is the litter daily weight gain.
Comment: Thank you for your comment. The adjustments were made in the new version. The value refers to the litter daily weight gain.
Eq. 1: Line 207: What is "(piglet)" after weight gain? Is this the number of piglets? If you weighed all the individual piglets, why not just use the sum of their birth weights rather than an average birth weight ("initial body weight (kg piglet -1)") x the number of piglets?
Comment: Thank you for your comment. As the proposal to facilitate the understanding of the equation, corrections and adjustments were made in the new version.
- Lines 209-212: Is there a reason you chose the weight categories you did for birth and weaning?
Comment: Thank you for your comment. No. We try to categorize the piglets in the litter by fixed weight ranges.
- Lines 216-242: FCR and ADFI had to be calculated on a per pen basis (4 boars/pen according to lines 108-109). Weights were recorded on individual pigs but the treatment is applied to the entire pen. Was ADG during the nursery analyzed as a repeated measures with pen as the subject or was it analyzed on the group of 4 pigs? This is unclear and would be the only way to properly analyze ADG.
Comment: Thank you for your comment. The adjustments were made in the new version. The average daily weight gain (ADG) was calculated by subtracting the piglet weight from the previous weighing phase and adjusted to the arithmetic mean in each experimental unit.
- Line 231: missing a space - p< 0.05 instead of p <0.05
Comment: Thank you for your comment. The adjustments were made in the new version.
- Tables 5-8: There are several spots where a "," is used instead of a "." to separate the whole number from the tenths/hundredths places.
Comment: Thank you for your comment. The adjustments were made in the new version.
- Table 5: Consider including d35 with initial body weight as a reminder that the initial body weight of the sow was recorded on d35 of gestation.
Comment: Thank you for your comment. The adjustments were made in the new version.
- Also, regarding sow weights: did you consider using the equations by Noblet et al., 1985 (Br. J. Nutr. 53:251-265) to adjust the sow weight for fetal mass and milk production?
Comment: Thank you for your comment. The equations cited were not used to adjust sow weight to fetal mass and milk production. The weight of the sows was used as a blocking factor, which was essential to equalize possible biased analyses.
- Table 6: Total born and live born under Control have only 1 number behind the decimal, while the others have 2. Please be consistent in the usage of decimal places. Same for p-value for Average birth interval. All your other p-values are to 3 decimal places.
Comment: Thank you for your comment. The adjustments were made in the new version.
- Table 6, 7: Did you consider analyzing the number of piglets in each weight category using litter size as a covariate instead of analyzing the percentage of piglets in each weight category?
Comment: Thank you for your comment. For the litter size variable, the number of piglets born in the previous cycle and the parity order were used as a blocking factor, which was essential to equalize possible biased analyses. We understand that the size of the litter at birth is a variable response to the gestation phase, therefore not desirable to be controlled.
- Table 8: Body weight - initial = have a superscript on Average of 4 which has a footnote that states that the Means differ significantly. However, if I read your table correctly, the difference between these two values has a p-value of 0.9. I assume this should be a superscript of 6 with a footnote that states "Average factor: supplementation in nursery" similar to superscript 2. Also, in the footnotes, superscript 5 is not in superscript format. I would also consider using "sow" instead of "reproduction" for this table.
Comment: Thank you for your comment. The adjustments were made in the new version.
- Lines 328-329: Wording could be better. Also, need citation on blood flow to the uterine horns. You state "typically result in lighter piglets" but reference 33 says it does and you don't have any other citations (i.e. the manuscripts 33 references as having shown differing effects of uterine horn position on birth weight).
Comment: Thank you for your comment. The adjustments were made in the new version.
- Abbreviations: You have abbreviations listed here that I didn't see used in the manuscript and abbreviations that were used in the manuscript that are missing from this list.
Comment: Thank you for your comment. The adjustments were made in the new version.
